

# Development and validation of ferroptosis-related lncRNAs signature for hepatocellular carcinoma

Jiaying Liang[1,2,*], Yaofeng Zhi[1,2,*], Wenhui Deng[3], Weige Zhou[1], Xuejun Li[1], Zheyou Cai[1], Zhijian Zhu[1], Jinxiang Zeng[1], Wanlan Wu[1], Ying Dong[1], Jin Huang[4], Yuzhuo Zhang[1,2], Shichao Xu[1,2], Yixin Feng[1,2], Fuping Ding[5] and Jin Zhang[1,2]

[1] Guangzhou University of Chinese Medicine, School of Basic Medical Sciences, Guangzhou, China
[2] Guangzhou University of Chinese Medicine, Research Center of Integrative Medicine, School of Basic Medical Sciences, Guangzhou, China
[3] Guangzhou University of Chinese Medicine, The fourth Affiliated Hospital of Guangzhou University of Chinese Medicine, Shenzhen, China
[4] Guangzhou University of Chinese Medicine, Clinic of Guangzhou University of Chinese Medicine, Guangzhou, China
[5] Guangzhou University of Chinese Medicine, School of Nursing, Guangzhou, China
* These authors contributed equally to this work.

Corresponding authors
Fuping Ding, hldfp@gzucm.edu.cn
Jin Zhang, zhjin@gzucm.edu.cn

## ABSTRACT

**Background**. Hepatocellular carcinoma (HCC) with high heterogeneity is one of the most frequent malignant tumors throughout the world. However, there is no research to establish a ferroptosis-related lncRNAs (FRlncRNAs) signature for the patients with HCC. Therefore, this study was designed to establish a novel FRlncRNAs signature to predict the survival of patients with HCC.

**Method**. The expression profiles of lncRNAs were acquired from The Cancer Genome Atlas (TCGA) and Gene Expression Omnibus (GEO) database. FRlncRNAs co-expressed with ferroptosis-related genes were utilized to establish a signature. Cox regression was used to construct a novel three FRlncRNAs signature in the TCGA cohort, which was verified in the GEO validation cohort.

**Results**. Three differently expressed FRlncRNAs significantly associated with prognosis of HCC were identified, which composed a novel FRlncRNAs signature. According to the FRlncRNAs signature, the patients with HCC could be divided into low- and high-risk groups. Patients with HCC in the high-risk group displayed shorter overall survival (OS) contrasted with those in the low-risk group ($P < 0.001$ in TCGA cohort and $P = 0.045$ in GEO cohort). This signature could serve as a significantly independent predictor in Cox regression (multivariate HR > 1, $P < 0.001$), which was verified to a certain extent in the GEO cohort (univariate HR > 1, $P < 0.05$). Meanwhile, it was also a useful tool in predicting survival among each stratum of gender, age, grade, stage, and etiology,etc. This signature was connected with immune cell infiltration (i.e., Macrophage, Myeloid dendritic cell, and Neutrophil cell, etc.) and immune checkpoint blockade targets (PD-1, CTLA-4, and TIM-3).

**Conclusion**. The three FRlncRNAs might be potential therapeutic targets for patients, and their signature could be utilized for prognostic prediction in HCC.

# INTRODUCTION

Hepatocellular carcinoma (HCC), which is the second frequent cause of death in human cancers throughout the world, is one of the most common malignant tumors (*Llovet et al., 2016*). It was estimated that approximately 841,000 new cases of HCC are diagnosed annually and approximately 781,631 patients would die of HCC in 2018 (*Bray et al., 2018*). For early-stage patients, radiofrequency local ablation, partial hepatectomy and liver transplantation are the major therapies and about 70% of patients will relapse within five years after operation (*European Association for the Study of the Liver, 2018*). Immune checkpoint inhibitors have been proven to be effective strategies for the treatment of advanced HCC, but their effectiveness still need to be further improved (*Yang et al., 2019*). Despite the advances in early detection, and drug development, the clinical outcomes of advanced cases remain unsatisfactory. The 5-year survival rate of local HCC is 30.5%, and that of distant metastasis was less than 5% (*Oweira et al., 2017*). To improve clinical outcomes and reduce the burden of cases, it is urgent to identify novel effective molecular markers and ameliorate prediction of HCC prognosis.

Ferroptosis is an iron-dependent modality of regulated cell death driven by the malignant accumulation of lipid peroxidation (*Dixon et al., 2012*; *Stockwell et al., 2017*). Recently, the induction of ferroptosis has been listed as a promising therapeutic strategy, especially suitable for malignant tumors that respond to resistance in traditional treatments (*Hassannia, Vandenabeele & Vanden Berghe, 2019*; *Liang et al., 2019*). A large number of experimental studies had indicated that ferroptosis-related genes played a vital role in HCC (*Jennis et al., 2016*; *Louandre et al., 2015*; *Sun et al., 2016a*; *Sun et al., 2016b*; *Yuan et al., 2016*).

Long non-coding RNA (lncRNA) with a minimum length of about 200 nucleotides are autonomous transcriptional RNA which does not encode proteins (*Cech & Steitz, 2014*). LncRNAs have been proven to be abnormally expressed in multiple cancers, and aberrant lncRNAs have been reported to serve as prognostic indicators in various cancers including HCC (*Ai et al., 2020*; *He et al., 2019*; *Li et al., 2019*; *Wang et al., 2018*; *Ye et al., 2019*; *Zeng et al., 2020*; *Zhao, Liu & Yu, 2017*). One recent study revealed that ferroptosis-related lncRNAs signature was associated with the prognosis of patients with head and neck squamous cell carcinoma (*Tang et al., 2021*). However, there is little research on ferroptosis-related lncRNAs correlated with HCC patient prognosis. Therefore, this study aims to establish a novel ferroptosis-related lncRNAs (FRlncRNAs) signature in predicting the prognosis of patients with HCC, hoping to improve current diagnosis, treatment, follow-up and prevention strategies of HCC.

## MATERIALS AND METHODS

### Data source and clinical information

The RNA sequencing (RNA-seq) data together with relevant clinical data were accessed from The Cancer Genome Atlas (TCGA) database (https://portal.gdc.cancer.gov/). An overview of the clinical information and source file of the patients with HCC can be found in Table 1 and Table S1. Notably, those patients with follow-up time greater than one month were used for the study. Totally, 374 Hepatocellular carcinoma (HCC) patients were available for further analysis. The GSE14520 dataset was acquired from GEO database (http://www.ncbi.nlm.nih.gov/geo/), containing 488 patients with HCC. A total of 259 ferroptosis-related genes (Marker: 111; Driver: 108; Suppressor: 69) were identified from FerrDb Database (*Zhou & Bao, 2020*) (FerrDb, http://www.zhounan.org/ferrdb/; Table S2). The TCGA dataset was utilized for training cohort while the GSE14520 dataset was validation cohort.

### LncRNAs and ferroptosis-related genes data processing

The "limma" package was employed to select differentially expressed ferroptosis-related genes, which were visualized through the volcano and heatmaps. Then, we carried out functional enrichment analysis (Gene Ontology (GO) and Kyoto Encyclopedia of Genes and Genomes (KEGG)) to determine the major biological attributes. The "GOplot" package was utilized to visualize enrichment terms.

To calculate the correlation between candidate ferroptosis-related lncRNAs (FRlncRNAs) and differentially expressed ferroptosis-related genes utilizing Pearson correlation. The coefficient $P < 0.001$ and $|R^2| > 0.3$ were regarded to be FRlncRNAs. Finally, Cytoscape software was utilized to draw co-expression network of prognostic FRlncRNAs and ferroptosis-related genes.

### Construction of prognostic FRlncRNAs signature

First, the FRlncRNAs associated with prognosis were assessed using univariate Cox regression in training cohort. Then, FRlncRNAs with $P \leq 0.05$ were included into multivariate Cox regression for the construction of FRlncRNAs signature. The formula utilized was as follows: risk score of FRlncRNAs signature $= \sum_i Coefficient\,(FRlncRNAs_i) * Expression(FRlncRNAs_i)$. To stratify patients into low- or high-risk groups, the best cut-off of the FRlncRNAs signature was identified applying receiver operating characteristic (ROC) curve at 1 year for overall survival (OS). Survival analysis between the two risk groups were assessed by Kaplan–Meier (KM) and compared using log-rank statistical methods.

Nomogram was utilized to predict 1, 3 years, and 5 years survival for the patients with HCC. ROC curves and Calibration curves were utilized to explore the accuracy of model based on training cohort. Then, we adjusted other clinical features in independent prognostic analysis in order to confirm whether the FRlncRNAs signature was an independent indicator to predict the prognosis of patient with HCC.

### Validation of the FRlncRNAs signature

The GEO cohort was enrolled to verify the robustness of model established from training cohort. The FRlncRNAs signature was calculated based on validation cohort. Then, survival

**Table 1  Clinical characteristics of the patients with HCC from the TCGA cohort in this study.**

| Variables | Number of patients | Percent (%) |
|---|---|---|
| **Age** | | |
| <65 | 224 | 59.42 |
| ≥65 | 152 | 40.32 |
| NA | 1 | 0.27 |
| **Gender** | | |
| Male | 255 | 67.64 |
| Female | 122 | 32.36 |
| **Grade** | | |
| G1 | 55 | 14.59 |
| G2 | 180 | 47.75 |
| G3 | 124 | 32.89 |
| G4 | 13 | 3.45 |
| NA | 5 | 1.33 |
| **Stage** | | |
| stage I | 175 | 46.42 |
| stage II | 88 | 23.34 |
| stage III | 86 | 22.81 |
| stage IV | 6 | 1.59 |
| NA | 22 | 5.84 |
| **Etiology** | | |
| Alcohol Liver Disease | 118 | 31.30 |
| NAFLD | 12 | 3.18 |
| HBV/HCV | 118 | 31.30 |
| Hemochromatosis | 5 | 1.33 |
| NA | 124 | 32.89 |
| **Radiotherapy** | | |
| YES | 8 | 2.12 |
| NO | 338 | 89.66 |
| NA | 31 | 8.22 |
| **Family history** | | |
| YES | 114 | 30.24 |
| NO | 212 | 56.23 |
| NA | 51 | 13.53 |

**Notes.**
NAFLD, nonalcoholic fatty liver disease; HBV, hepatitis B virus; HCV, hepatitis C virus.

analysis and Cox regression were utilized to evaluate whether the FRlncRNAs signature was significantly connected with OS in validation cohort. The ROC curves were established to assess whether the novel model could accurately predict patient survival.

## Gene set enrichment analysis

Gene set enrichment analysis (GSEA, http://www.broadinstitute.org/gsea/index.jsp) was utilized to investigate functional phenotypes differences between the two risk groups (high- and low-risk groups). In this research, we carried out functional enrichment of FRlncRNAs

signature, and visualized the pathway closely related to immune and tumorigenesis development. The reference gene sets contained "c7.all.v7.2.symbols.gmt [Immunologic signatures] and h.all.v7.2.symbols.gmt [cancer hallmarks]".

## Immune correlation analysis

The CIBERSORT (*Charoentong et al., 2017*; *Newman et al., 2015*), EPIC (*Racle et al., 2017*), ESTIMATE (*Yoshihara et al., 2013*), MCP counter (*Shi et al., 2020*), QUANTISEQ (*Finotello et al., 2019*), TIMER (*Li et al., 2017*), and single-sample gene set enrichment analysis (ssGSEA) (*Yi et al., 2020*) algorithms were used to infer the relative content of tumor-infiltrating immune cells (TIICs) between two risk group based on FRlncRNAs signature. The heatmap was utilized to visualize the differences of TIICs abundance under different algorithms. Besides, the correlation analysis between the abundance of TIICs and FRlncRNAs signature was utilized to reveal the potential role of FRlncRNAs signature on the immunologic features based on TIMER results.

The expression of immune checkpoint gene might be related to treatment responses of immune checkpoint inhibitors (ICIs) (*Goodman, Patel & Kurzrock, 2017*). Thus, we investigated six ICIs: programmed death 1 (PD-1) and its ligand 1 (PD-L1), ligand 2 (PD-L2), indoleamine 2,3-dioxygenase 1 (IDO1), cytotoxic T-lymphocyte antigen 4 (CTLA-4), and T cell immunoglobulin and mucin domain-containing protein-3 (TIM-3) in HCC (*Kim et al., 2017*; *Nishino et al., 2017*; *Zhai et al., 2018*). We analyzed the Spearman correlation between the ICIs and the signature, which aimed to investigate the potential role of FRlncRNAs signature in immune checkpoint blockade therapy.

## Statistical analysis

All the statistical analyses were conducted using R language (version 4.0). KM analysis with log-rank test from "survival" package was applied to compare the survival difference among two risk groups. In order to evaluate the prognostic value of the novel FRlncRNAs signature, univariate and multivariate analyses were conducted through Cox proportional hazards regression model. Stratification analysis was implemented based on age ($\geq 65$ and $<65$ years), gender (male and female), stage (stage 1–2 and stage 3–4), grade (grade 1–2 and grade 3–4), etiology (Alcoholic liver disease and nonalcoholic fatty liver disease; HBV/HCV and non-HBV/HCV), Radiotherapy (receive radiotherapy and no receive radiotherapy), and family history (have family history of cancer and no family history). GSEA was applied to differentiate between two risk groups of functional annotations. Statistical tests were bilateral, with $P$ value $\leq 0.05$ indicated statistically significant differences.

# RESULTS

## Differentially expressed ferroptosis-related genes

After extracting the expression values of 259 ferroptosis-related genes in the patient with HCC, 69 up-regulated genes and 13 down-regulated genes were authenticated (FDR <0.05, log2FC >1; Table S3). The differentially expressed ferroptosis-related genes were visualized by volcano and heatmaps (Figs. 1A–1C). The GO enrichment revealed that these differentially expressed ferroptosis-related genes mainly participated in cellular

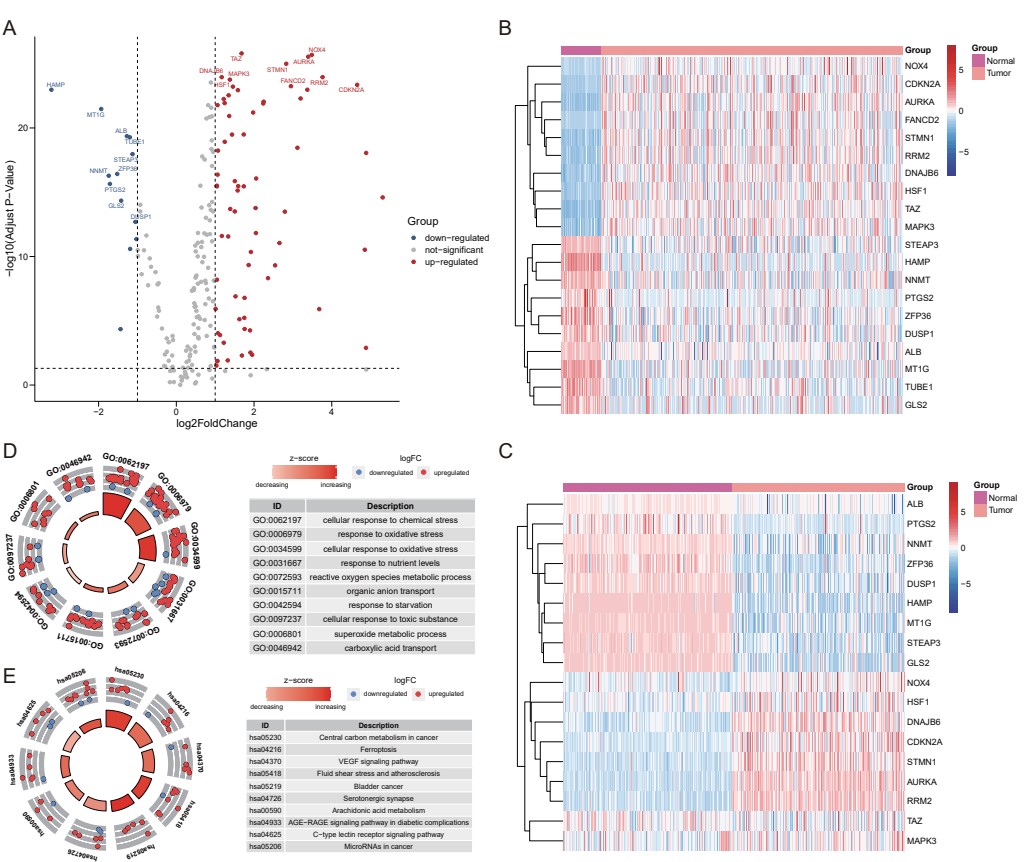

**Figure 1** **The differentially expressed ferroptosis-related genes.** (A) The Volcano plot of the differentially expressed ferroptosis-related genes. The red dots indicated up-regulated and blue for down-regulated. The top 20 gene symbols with high variation were displayed in the plot. (B) The heatmap of 20 ferroptosis-related genes with high variation in training cohort. (C) The heatmap of 20 ferroptosis-related genes with high variation (except for TUBE1, and FANCD2) between tumor and normal samples in validation cohort. (D) The GO circle plot of functional enrichment. The red dots indicated up-regulation, while the blue indicated down-regulation. (E) The KEGG circle plot of enrichment analysis. The Z-score was directly proportional to the level of enrichment.

response to chemical stress, response to oxidative stress, and carboxylic acid transport among others (Fig. 1D). The major KEGG pathways included Ferroptosis, VEGF signaling pathway, Arachidonic acid metabolism and some cancer-related signaling pathways (Fig. 1E; Table S4).

## Identification of prognostic FRlncRNAs and an FRlncRNAs signature

A total of 1,184 FRlncRNAs were identified in the training cohort through the correlation analysis between differentially expressed ferroptosis-related genes and lncRNAs ($|R^2|>0.3$ and $P < 0.001$; Table S5). Among them, 32 FRlncRNAs co-existed in training cohort and validation cohort were used for subsequent model construction and validation. The batch effects from different cohorts were corrected by combat function in "sva" package. Cox regression were applied to screen prognostic FRlncRNAs. In accordance with univariate cox results, eight FRlncRNAs had potential prognostic value for the patients with HCC

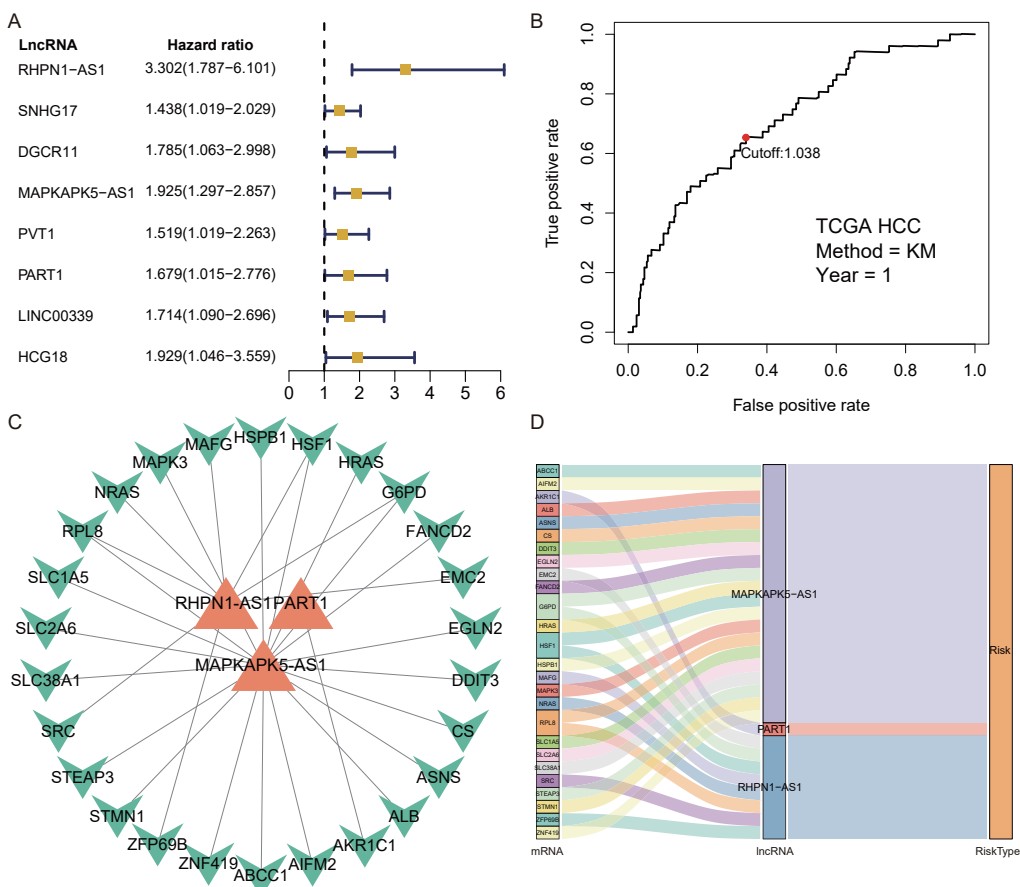

**Figure 2** **Identification of prognostic Ferroptosis-related lncRNAs (FRlncRNAs).** (A) Forest plots revealed prognosis-related FRlncRNAs based on the results of univariate Cox regression. (B) ROC curve for FRlncRNAs signature at 1 year in training cohort. The cut-off score was 1.038 and it was utilized to classify patients into high- or low-risk groups. (C) The relational network between signature and corresponding co-expression ferroptosis-related genes. (D) Sankey diagram indicated that the association between prognostic FRlncRNAs, ferroptosis-related genes, and risk type.

($P < 0.05$, Fig. 2A, Table S6). As shown in Table S6, eight FRlncRNAs (RHPN1 −AS1, SNHG17, DGCR11, MAPKAPK5 −AS1, PVT1, PART1, LINC00339 and HCG18) were discovered to be harmful prognostic indicators.

Subsequently, multivariate Cox regression found that three FRlncRNAs were associated with prognostic for the patients with HCC (Table S6). The three FRlncRNAs were utilized to establish a FRlncRNAs signature. The risk score was estimated using the following formula: risk score of FRlncRNAs signature = (0.89773*RHPN1-AS1) + (0.48558* MAPKAPK5-AS1) + (0.55674*PART1). The optimal cut-off point of risk score was considered to be 1.038 through ROC curve. Based on this cut-off point, the patients with HCC were classified into high- or low-risk group (Fig. 2B). The relationship between the three prognostic FRlncRNAs and co-expressed mRNA was shown in Fig. 2C and Fig. 2D. The risk score was significantly relevant to OS of patients, where OS in high-risk group possessed

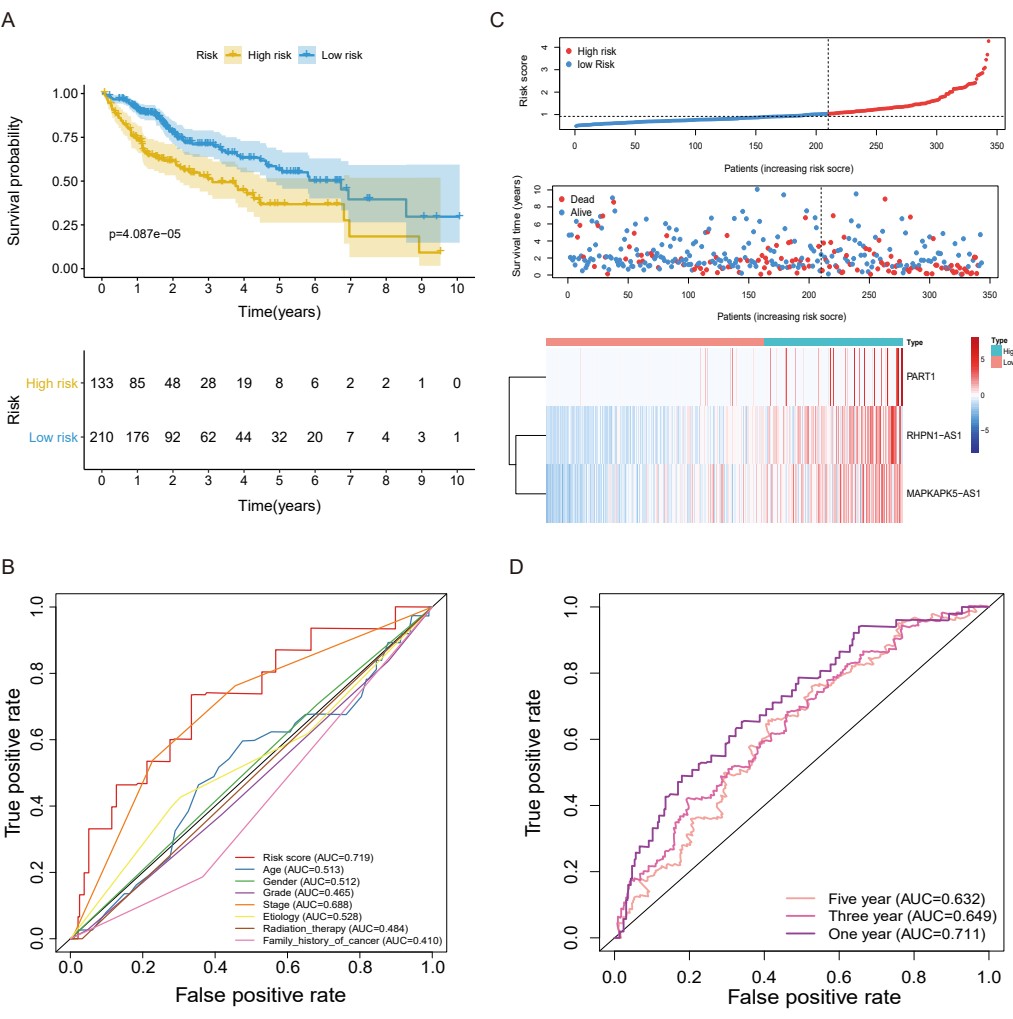

**Figure 3** **The FRlncRNAs signature based on training cohort.** (A) Kaplan–Meier (KM) curve for overall survival (OS) of patients with HCC in high- and low-risk group in training cohort. (B) The AUC values of the risk score and additional clinical characteristics. (C) Risk survival status plot in patients with HCC. (D) ROC curves at 1, 3, 5-year were applied to verify prognostic performance of FRlncRNAs signature established by training cohort.

shorter than those in low-risk group ($P < 0.001$, Fig. 3A). Concurrently, the AUC of the FRlncRNAs signature was 0.719, showing great performance in contrast to other traditional clinical pathological features in predicting the prognosis of patients with HCC (Fig. 3B). The survival status plot showed that risk score of patients was inversely proportional to their survival rate. Besides, the risk heatmap demonstrated that the expression of three FRlncRNAs had positive correlation with the risk levels (Fig. 3C). The areas under the ROC (AUC) values corresponding to 1, 3, and 5 years were 0.711, 0.649 and 0.632, respectively (Fig. 3D).

**Table 2  Univariate and multivariate independent prognostic analysis of FRlncRNAs signature in predicting patient survival.**

| Variables | Univariate Cox model | | Multivariate Cox model | |
|---|---|---|---|---|
| | HR(95% Cl) | p value | HR(95% Cl) | p value |
| Training cohort | | | | |
| Age | 1.014(0.997–1.032) | 0.104 | 1.012(0.994–1.031) | 0.181 |
| Gender | 0.701(0.457–1.075) | 0.103 | 0.912(0.555–1.500) | 0.718 |
| Grade | 1.035(0.778–1.376) | 0.815 | 0.971(0.707–1.333) | 0.856 |
| Stage | 1.853(1.464–2.345) | <0.001 | 1.807(1.412–2.313) | <0.001 |
| Etiology | 1.135(0.952–1.352) | 0.158 | 1.041(0.856–1.265) | 0.688 |
| Radiotherapy | 1.147(0.362–3.636) | 0.815 | 1.060(0.327–3.440) | 0.922 |
| Family history | 1.117(0.728–1.713) | 0.613 | 0.960(0.602–1.531) | 0.863 |
| Risk Score | 1.732(1.324–2.267) | <0.001 | 1.721(1.280–2.314) | <0.001 |
| Validation cohort | | | | |
| Gender | 1.689(0.816–3.497) | 0.158 | 1.381(0.659–2.891) | 0.392 |
| Age | 0.991(0.972–1.010) | 0.343 | 1.000(0.980–1.021) | 0.964 |
| TNM stage | 2.340(1.771–3.091) | <0.001 | 1.763(1.268–2.451) | <0.001 |
| CLIP stage | 1.921(1.554–2.375) | <0.001 | 1.493(1.155–1.930) | 0.002 |
| Risk Score | 1.410(1.054–1.887) | 0.021 | 1.049(0.764–1.439) | 0.769 |

## The FRlncRNAs signature was an independent prognostic indicator

Univariate independent prognostic analysis revealed that risk score was a prognostic factor and significantly associated with worse survival (HR = 1.732, 95% CI [1.324–2.267]; $P < 0.001$) (Table 2, Fig. 4A). Moreover, after adjusting other available clinical parameters such as age, gender, stage, grade, etiology, radiotherapy, and family history, our signature still maintained an independent prognostic factor in multivariate independent analysis (HR = 1.721, 95% CI [1.280–2.314]; $P < 0.001$) (Table 2, Fig. 4B). As indicated in the nomogram, three-FRlncRNA-based signature was the largest contribution to OS of each period in HCC (Fig. 4C). The calibration displayed that the FRlncRNAs signature possessed high accuracy (Figs. 4D–4F).

## Stratification analyses

In the training cohort, stratification analysis was conducted based on the clinicopathological features of HCC (e.g., gender, age, grade, stage, etiology, radiotherapy, and family history). As a result, the FRlncRNAs signature was still closely associated with worse survival in male, older (≥65 years) or younger (<65 years), advanced grade (Grade 3–4) or early grade (Grade 1–2) and advanced stage (Stage 3–4) or early stage (Stage 1–2) patients (all $P < 0.05$; Figs. 5A–5F), which indicated that FRlncRNAs signature based on risk grouping could serve as a useful tool for predicting HCC survival among each stratum of gender, age, grade, and stage. Meanwhile, this signature can also be utilized as a potential prognostic tool for HCC patients with alcoholic or non-alcoholic liver disease, no family history of cancer, HBV-positive, HBV- and HCV- negative, and no radiotherapy (Figs. 5G–5M).

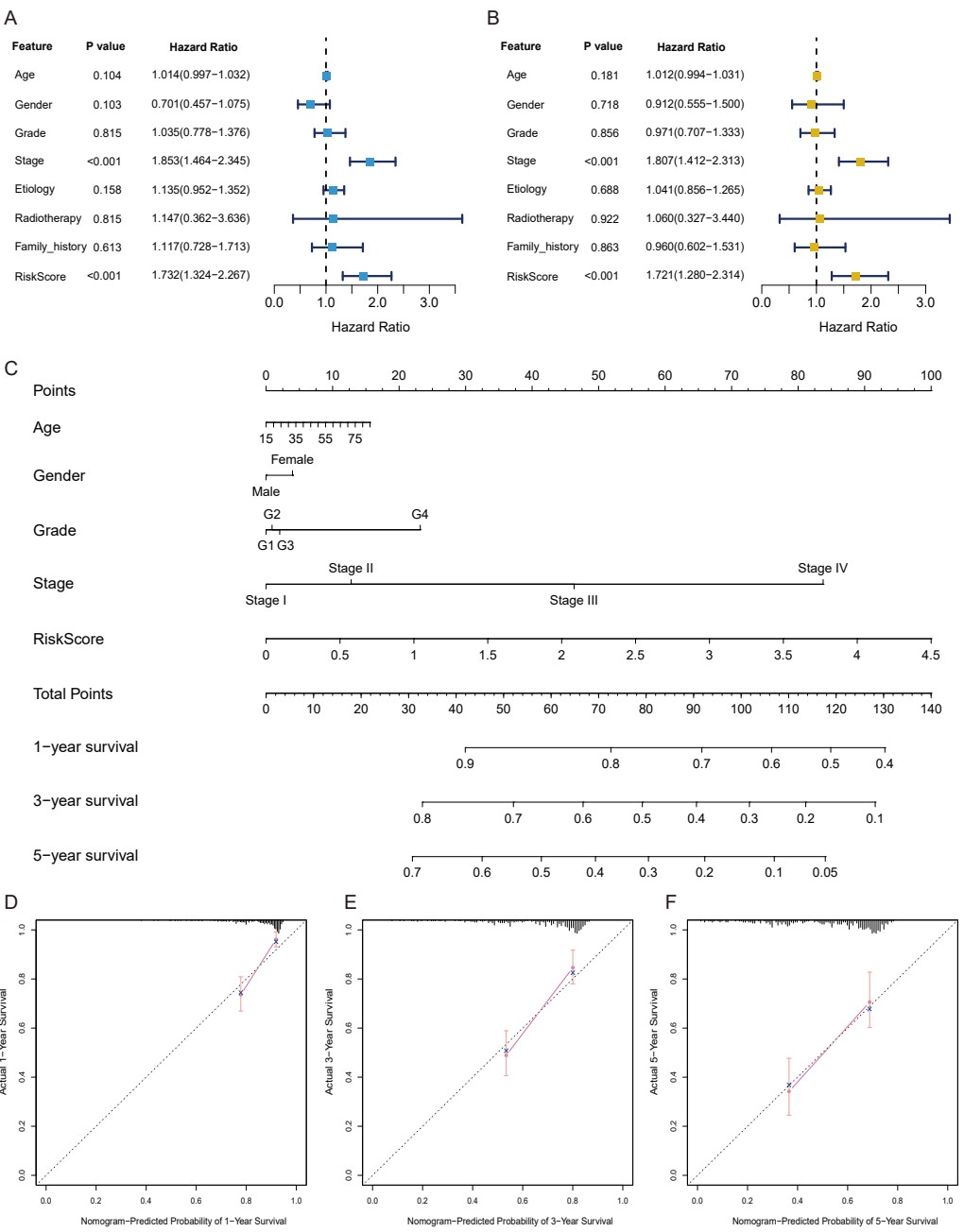

**Figure 4** **The FRlncRNAs signature was an independent prognostic indicator and possessed potential clinical value in the training cohort.** Univariate (A) and multivariate (B) Cox regression analysis of the FRlncRNAs signature in predicting patient survival. (C) A nomogram among clinical features (including the risk score) and OS of patients. (D–F) Calibration for assessing the consistency between the predicted and the actual OS at 1, 3, 5 years.

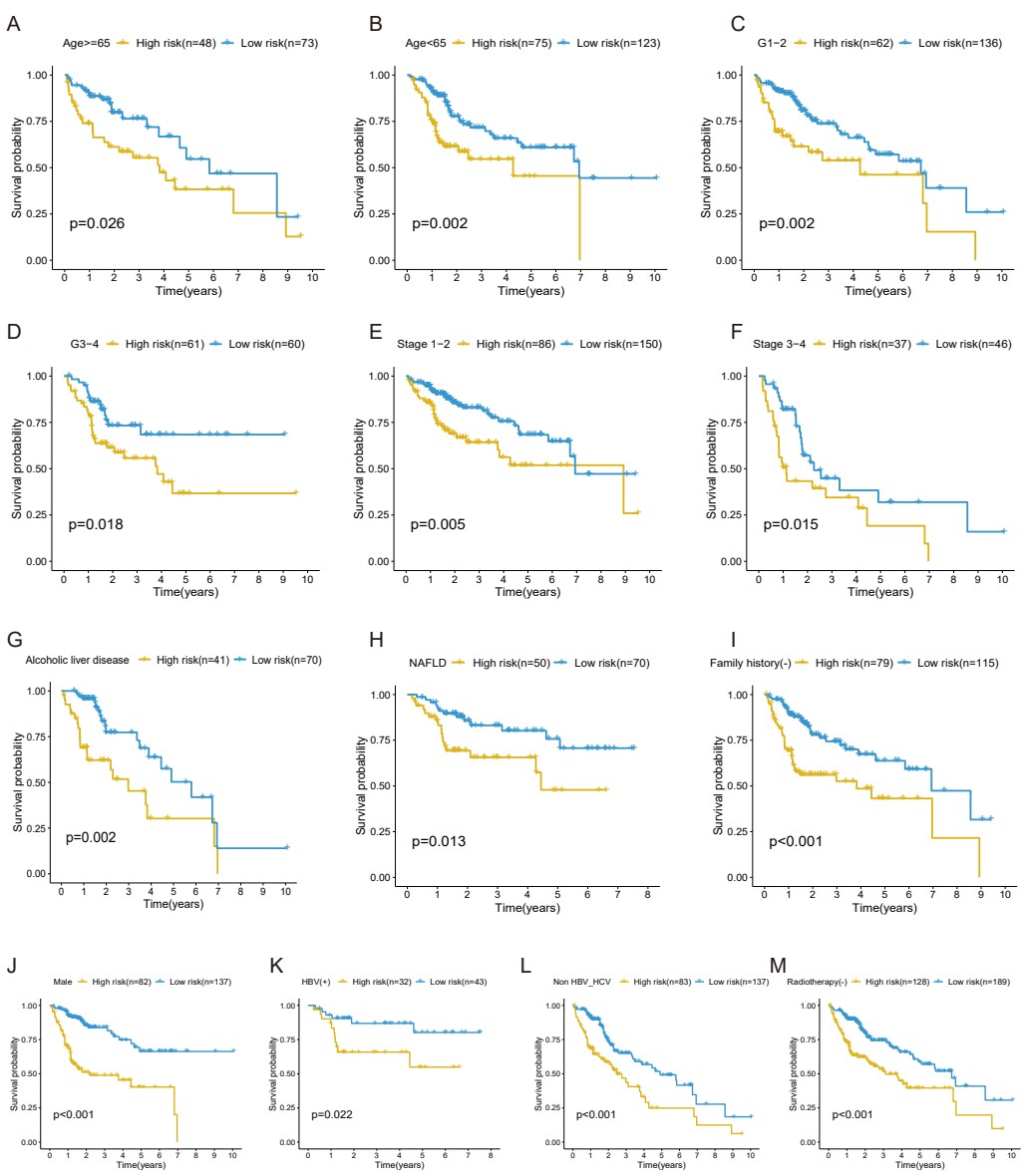

**Figure 5  The survival curves of the FRlncRNAs signature stratified by age, grade, stage, gender, etiology, radiotherapy, and family history.** (A) ≥ 65 years, (B) <65 years, (C) grade 1–2, (D) grade 3–4, (E) stage 1–2, (F) stage 3–4, (G) Alcoholic liver disease, (H) NAFLD, (I) no family history, (J) male, (K) HBV positive, (L) HBV and HCV negative, (M) no radiotherapy patients.

## Validation of the FRlncRNAs signature

In validation cohort, the risk score of FRlncRNAs signature was estimated refer to the previous formula. The cut-off point of the risk score of the validation cohort was consistent with that of the training cohort (Cutoff = 1.038). The signature was also statistically associated with OS of patients with HCC ($P = 4.504e{-}02$, Fig. 6A). Univariate independent prognostic analysis revealed that FRlncRNAs signature acted as an independent prognostic factor (HR of risk score = 1.410, 95% CI [1.054–1.887], $P < 0.05$, Table 2, Fig. 6B). Notably,

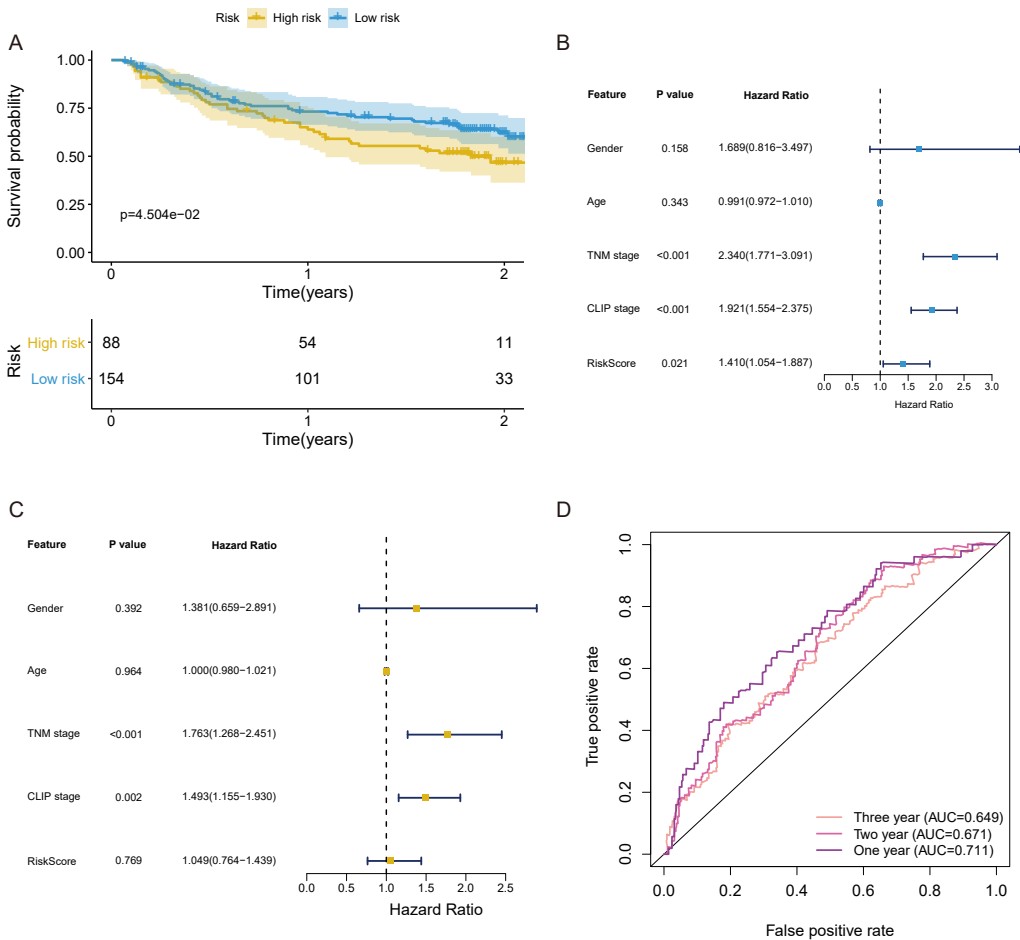

**Figure 6** **The verification of prognostic FRlncRNAs signature in validation cohort.** (A) The KM curves of validation cohort revealed that the high-risk group had statistical differences on OS period compared with the low-risk group (*P* < 0.05). Univariate (B) and multivariate (C) COX regression for the FRlncR-NAs signature established by training cohort. (D) AUC of ROC curves validated the predicted performance of signature in the validation cohort.

after controlling gender, age, TNM stage, and CLIP stage, FRlncRNAs signature was no longer a prognostic factor in multivariate analysis (HR = 1.049, 95% CI [0.764–1.439], *P* > 0.05), which indicated that more independent cohorts needed to be included for validation (Table 2, Fig. 6C). The AUC values were more than 0.65 at 1 year, 2 years and 3 years, which showed that the FRlncRNAs signature established from training cohort had powerful accuracy and robustness (Fig. 6D).

## The FRlncRNAs signature mediated DNA repair, glycolysis, MYC targets, P53 pathway and PI3K AKT mTOR signaling

GSEA was utilized to explore the potential biological mechanisms of the FRlncRNAs signature involved in HCC progression. The results of cancer hallmarks indicated that DNA repair, glycolysis, MYC targets, P53 pathway and PI3K AKT mTOR signaling were

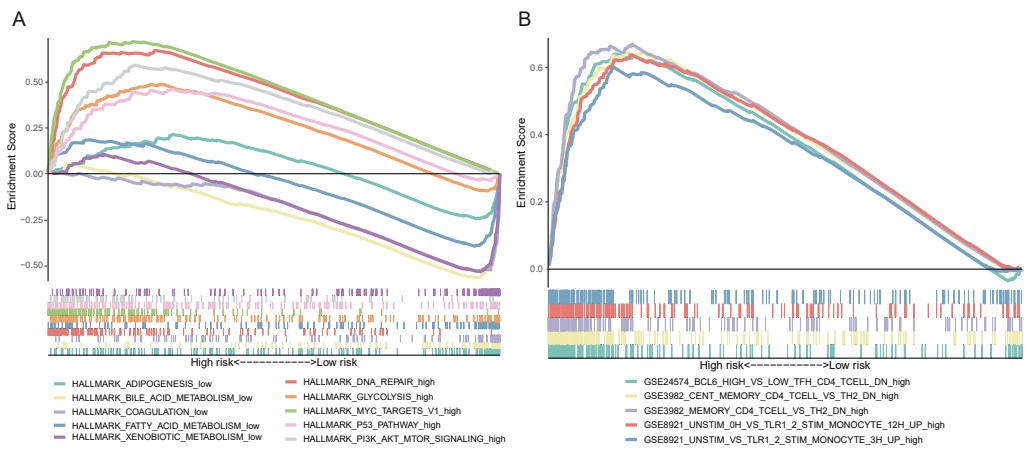

**Figure 7 Some cancer-related hallmarks and immunologic features regulated through the FRlncRNAs signature.** (A) Cancer hallmarks. (B) Immunologic signatures.

activated by the high-risk group of the FRlncRNAs signature. While adipogenesis, bile acid metabolism, coagulation, fatty acid metabolism and xenobiotic metabolism were activated by the low-risk group (Fig. 7A). Moreover, the signature also regulated many immunologic features within immune system, such as BCL6 high versus low TFH CD4 T cell down, cent memory CD4 T cell versus TH2 down etc., which indicated that the FRlncRNAs signature was implicated in immunity-related regulation (Fig. 7B).

## Correlation of FRlncRNAs signature with TICIs and immune checkpoint blockade (ICB) molecule

Many immunologic features were found to be regulated by FRlncRNAs signature according to the above results. Therefore, the signature was further explored whether it was associated with TIICs based on TIMER results. The results indicated that this signature was most significantly positive correlation with immune infiltration of Neutrophil cells (COR = 0.312, $P < 0.001$), Myeloid cells (COR = 0.268, $P < 0.001$), CD4+ T cells (COR = 0.203, $P < 0.001$), B cells (COR = 0.130, $P = 0.016$), and Macrophage cells (COR = 0.124, $P = 0.021$; Figs. 8A–8F). The heatmap of immune responses based on CIBERSORT, EPIC, ESTIMATE, MCP counter, QUANTISEQ, TIMER and ssGSEA algorithms was displayed in Fig. 9. These findings powerfully indicated that this FRlncRNAs signature was related to immune cell infiltration in HCC.

Tumor immunotherapy utilizing ICB had gradually become a promising strategy for therapy of advanced HCC (*Sangro et al., 2021*; *Wing-Sum Cheu & Chak-Lui Wong, 2021*; *Zongyi & Xiaowu, 2020*). In training cohort, we carried out the association between the FRlncRNAs signature and six common ICB therapy-related targets (PD-1, PD-L1, PD-L2, TIM-3, IDO1, and CTLA-4) to explore the potential role of FRlncRNAs signature in the immunotherapy of ICB in the patients with HCC. The results showed that the FRlncRNAs signature was positively related to PD $-1$ ($R = 0.17$, $P = 0.0019$), CTLA-4 ($R = 0.19$, $P = <0.001$), and TIM-3 ($R = 0.16$, $P < 0.001$; except for IDO1, PD-L1, and PD-L2),

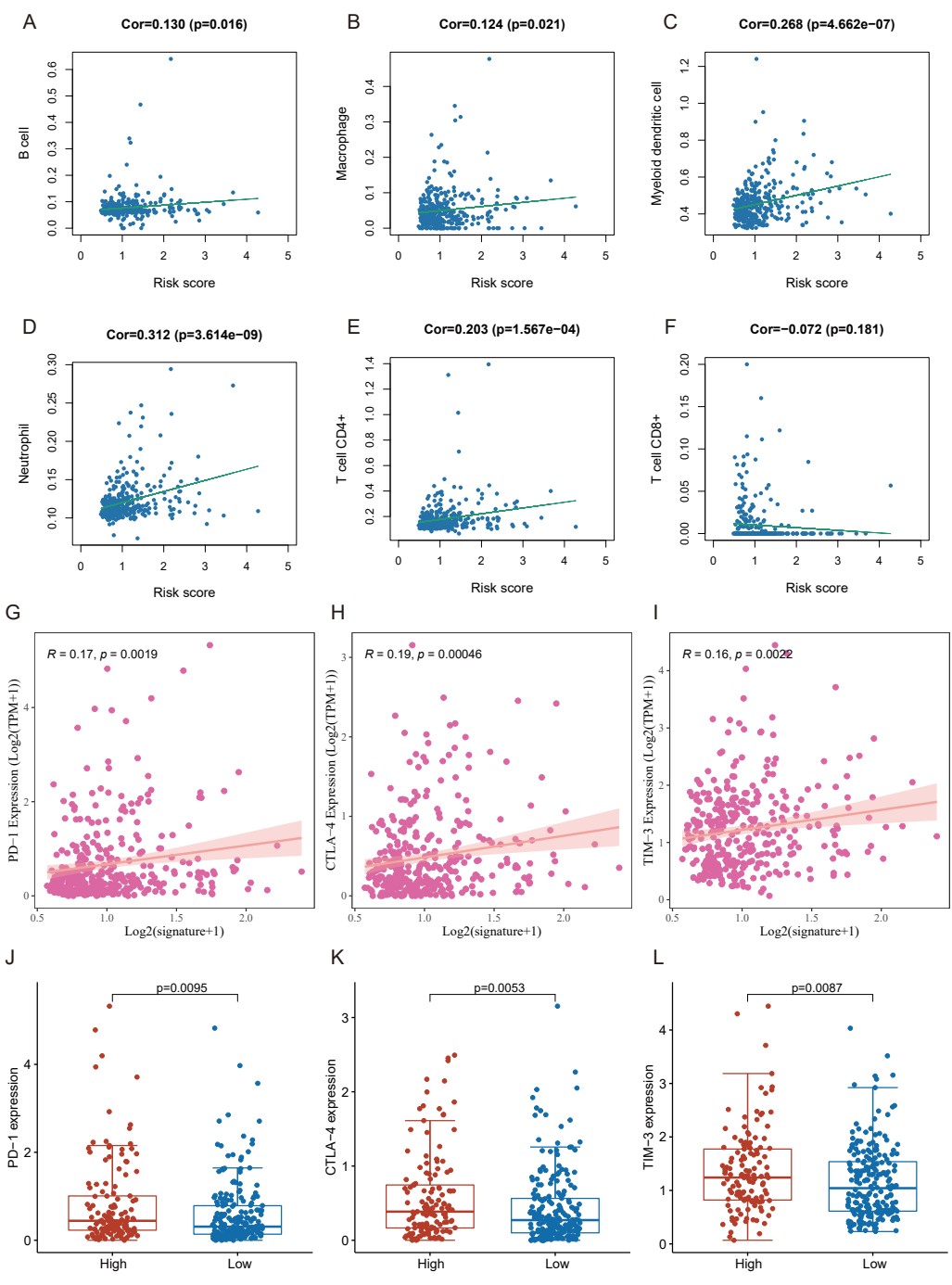

**Figure 8** **The relationship between FRlncRNAs signature and TIICs, ICB molecules based on TIMER results.** (A) Spearman correlation between the signature and B cell; (B) Spearman correlation between the signature and Macrophage cell. (C) Spearman correlation between the signature and Myeloid dendritic cell. (D) Spearman correlation between the signature and Neutrophil cell. (E) Spearman correlation between the signature and CD4+ T cell. (F) Spearman correlation between the signature and CD8+ T cell. (G–I) Significant positive association between our FRlncRNAs signature and ICB receptors PD-1 ($R$ = 0.17; P = 0.0019), CTLA-4 ($R$ = 0.19; P <0.001), and TIM-3 ($R$ = 0.16; P < 0.001). (J–L) The comparison of the expression levels of PD-1, CTLA-4, and TIM-3 between high-risk and low- groups.

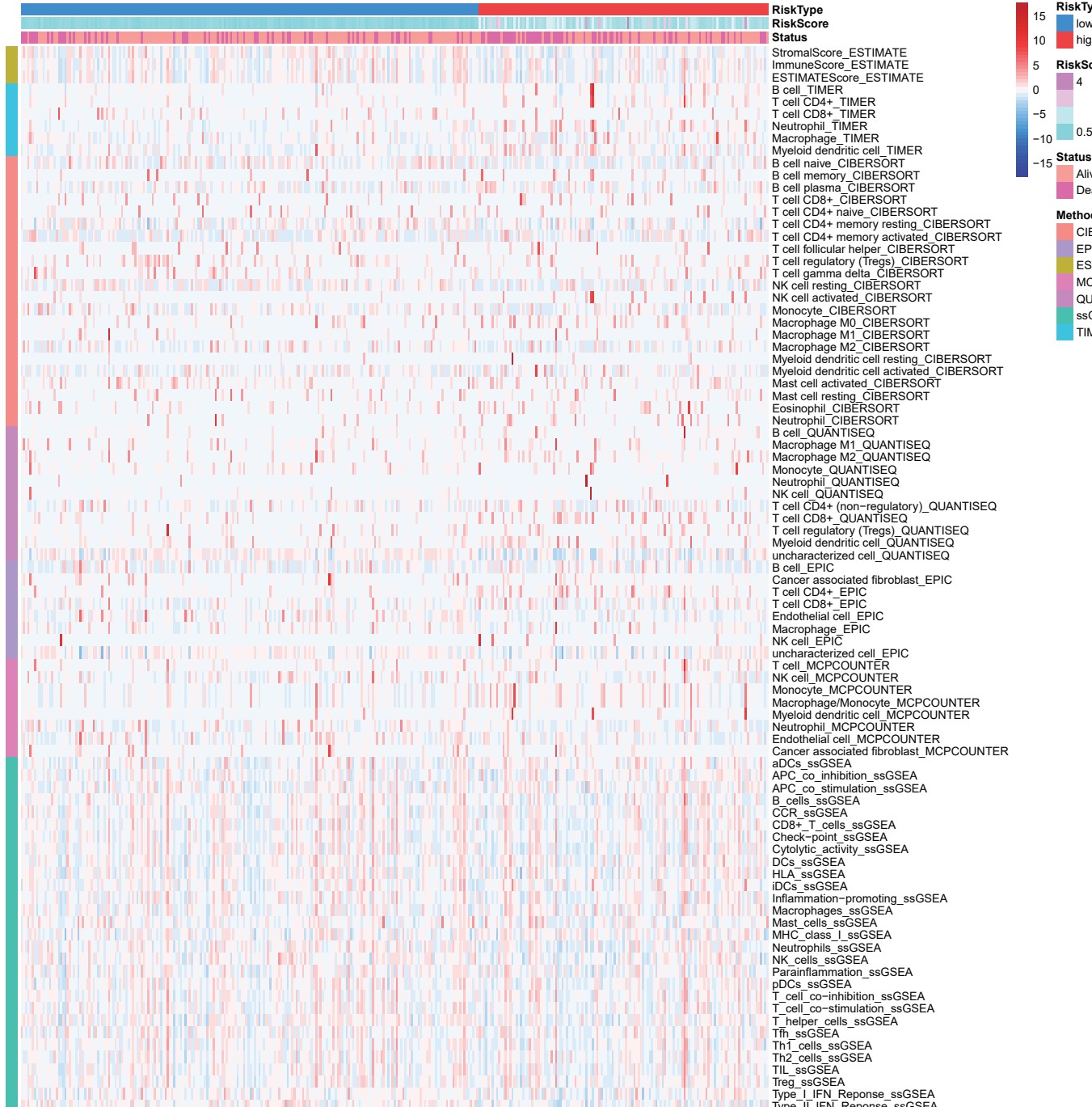

**Figure 9** Based on CIBERSORT, EPIC, ESTIMATE, MCP counter, QUANTISEQ, ssGSEA and TIMER algorithms, heatmap of immune infiltration in the high- and low-risk groups.

revealing that the FRlncRNAs signature might play vital roles in assessment of response to ICB immunotherapy in the patients with HCC (Figs. 8G–8I and Fig. S1). Meanwhile, the expression levels of PD-1, CTLA-4 and TIM-3 were significantly higher in high-risk group contrasted with those in low- group (Figs. 8J–8L).

## DISCUSSION

Due to the unique molecular features such as genomic and genetic diversities, HCC was considered as a highly heterogeneous malignant tumor (*Cancer Genome Atlas Research Network. Electronic address wbe, and Cancer Genome Atlas Research N, 2017*; (*Schulze, Nault & Villanueva, 2016*). Studies found that lncRNAs play important roles in the prognosis of HCC, which could become potential and effective molecular targets in the treatment of HCC (*DiStefano, 2017*; *Wei et al., 2019*). It could be seen from previous studies that lncRNAs participate in many biological processes such as immune response, autophagy, inflammation, and metabolism, etc (*Carpenter & Fitzgerald, 2018*; *Frankel, Lubas & Lund, 2017*; *Majidinia & Yousefi, 2016*; *Mathy & Chen, 2017*). At present, emerging studies revealed that some lncRNAs could play significant roles in regulating occurrence and development of disease by promoting ferroptosis (*Lu, Xu & Lu, 2020*; *Wang et al., 2021b*; *Yang et al., 2020b*). That indicated that ferroptosis-related lncRNAs might serve as novel disease molecular biomarkers and therapeutic targets for the treatment of cancer. However, the possible role of the ferroptosis-related lncRNA signature as a potentially useful tactics of treatment have not been reported in HCC. Therefore, we developed a FRlncRNAs signature with great prognosis and predictive value. Meanwhile, we also explored its effect in the response to ICB therapy for the patients with HCC.

Currently, the ferroptosis-related lncRNA signature has only been explored in head and neck squamous cell carcinoma (*Tang et al., 2021*). However, that study had several limitations, such as: small clinical sample sizes and lack of independent external validation datasets, which might lead to unreliable results. Furthermore, other confounding factors like comorbidities or alcohol consumption may also affect the robustness and accuracy of signature, which contained hemochromatosis, alcoholic liver disease, and nonalcoholic fatty liver disease, etc (*Mao et al., 2020*). Considering the above problems, the training cohort and independent validation cohort were utilized to develop a novel FRlncRNAs signature, which could predict the survival of patients with HCC. The results showed that the FRlncRNAs signature mainly involved DNA repair, glycolysis, MYC targets, and tumor-related signaling pathways. Those patients in the high-risk group were associated with worse OS. In addition, this FRlncRNAs signature was credible for predicting the prognosis of patients with HCC, with a high AUC value (average AUC >0.65), and might serve as an indicator to measure the response of patients with HCC to ICB immunotherapy.

With the development of immune checkpoint inhibitors, ICB immunotherapy, as emerging strategies, has revealed treatment effects in HCC (*Llovet et al., 2018*; *Pitt et al., 2016*). Currently, immunotherapy has provided a novel promising treatment strategy for HCC (*Sim & Knox, 2018*). Unfortunately, more than two-third of patients did not respond to ICB treatment (*Mushtaq et al., 2018*). A new research demonstrated that ferroptosis

combined with ICIs could synergistically enhance anti-tumor activity, even in ICI-resistant tumors (*Tang et al., 2020*). Thus, a novel FRlncRNAs signature was established to investigate the relationship between ICIs and ferroptosis, and predict ICB immunotherapy responses. In our study, the FRncRNAs signature was discovered to be associated with ICIs (i.e., PD-1, CTLA-4, and TIM-3), which indicated that the FRlncRNAs signature have potential to be used to measure the response to ICB therapy. At the same time, the expression levels of these ICIs in high-risk group were higher compared with low- group. That indicated that the FRlncRNAs signature could be applied to predict the expression level of ICIs and have the potential to guide ICB immunotherapy strategies. Moreover, the FRlncRNAs signature was connected with TICIs (B cell, Macrophage, Myeloid dendritic cell, Neutrophil, and CD4+ T cell) in HCC, which implies that this signature may play an important role in immune infiltration. Notably, these findings were consistent with previous studies manifesting that several lncRNAs served as regulators in tumor immunity, for instance immune cell infiltration and antigen release (*Carpenter & Fitzgerald, 2018*; *Denaro, Merlano & Lo Nigro, 2019*).

Previous studies have revealed that lncRNAs participated in different biological processes such as immune regulation (*Denaro, Merlano & Lo Nigro, 2019*), DNA repair and cell cycle (*Hu et al., 2018*; *Majidinia & Yousefi, 2016*), and metabolism (*Denaro, Merlano & Lo Nigro, 2019*), etc. Among this FRlncRNAs signature (RHPN1-AS1, MAPKAPK5-AS1, and PART1) of our study, RHPN1-AS1 could facilitate cell proliferation, and invasion via activating PI3K/AKT/mTOR pathway in HCC (*Song et al., 2020*). Another study revealed that RHPN1-AS1 promoted the progression of HCC through regulating miR-596/IGF2BP2 axis (*Fen et al., 2020*). MAPKAPK5-AS1/PLAGL2/HIF-1 α signaling pathway was found to drive the progression of HCC and MAPKAPK5-AS1 might be a novel therapeutic target (*Wang et al., 2021a*). Furthermore, MAPKAPK5-AS1 has been discovered to promote the progression of colorectal cancer and thyroid cancer (*Ji et al., 2019*; *Yang et al., 2020a*; *Zhou et al., 2020b*). PART1 was involved in cell migration and invasion, and it could facilitate progression of HCC (*Pu et al., 2020*; *Zhou et al., 2020a*). PART1 also played a vital role in the occurrence and development of other cancers. Downregulated PART1 could suppress proliferation and accelerate apoptosis in bladder cancer (*Hu et al., 2019*). PART1 was regarded as a novel target in the treatment of prostate cancer (*Sun et al., 2018*). A study found that PART1 could promote cell proliferation in non-small-cell lung cancer cells by targeting miR-17-5p (*Chen et al., 2021*). Above evidences revealed that these three FRlncRNAs played important roles in development and prognosis of HCC. However, there is no research on their role in the prognosis of HCC via ferroptosis-related mechanism. Our findings may provide a new perspective for the treatment of HCC through ferroptosis-induction in the future.

Additionally, our results showed that the new FRlncRNAs signature possessed a highly predictive ability for OS prediction in the patients with HCC. Stratification analysis indicated that the FRlncRNAs signature based on risk grouping still possessed great predictive ability for survival prediction in each stratum of age (<65 or ≥65 patients), stage (Stage 1–2 or Stage 3–4 patients), grade (Grade 1–2 or Grade 3–4), and male patients, etc.

Several issues remained in the current study. First, the clinical sample size was not large. Second, the prognostic model was demanded to be validated in other enormous datasets to guarantee its robustness. Third, study has shown that HCC could be resistant to conventional chemotherapeutic, which might be associated with induction of ferroptosis-resistance (*Galmiche, 2019*). However, due to the lack of patients of chemotherapy or radiotherapy in this study, it is impossible to confirm whether the signature can be applied to predict resistance to classical chemotherapy or radiotherapy through ferroptosis-induction. Fourth, the functional experiments should be implemented to reveal the potential biological mechanisms for predicting the influence of FRlncRNAs.

## CONCLUSIONS

In conclusion, we identified a novel FRncRNAs signature related to prognosis of the patients with HCC, which could be utilized as a powerful tool in predicting the prognosis of patients with HCC. The FRlncRNAs signature could divide clinical characteristic subgroups according to survival. In addition, the signature was connected with ICB targets and TICIs. Hence, our study afforded a possible strategy for individualized risk stratification of the patients with HCC and evaluation response to ICB immunotherapy. The three FRncRNAs might be potential therapeutic targets of HCC.

**Abbreviations**

| | |
|---|---|
| **HCC** | Hepatocellular carcinoma |
| **lncRNA** | Long non-coding RNA |
| **GO** | Gene Ontology |
| **KEGG** | Kyoto Encyclopedia of Genes and Genomes |
| **FRlncRNAs** | ferroptosis-related lncRNAs |
| **ROC** | receiver operating characteristic |
| **OS** | overall survival |
| **KM** | Kaplan–Meier |
| **GSEA** | Gene set enrichment analyses |
| **ssGSEA** | single-sample gene set enrichment analysis (ssGSEA) |
| **TIICs** | tumor-infiltrating immune cells |
| **ICB** | immune checkpoint blockade |
| **FDR** | false discovery rate |
| **NAFLD** | nonalcoholic fatty liver disease |
| **HBV** | hepatitis B virus |
| **HCV** | hepatitis C virus |

### Funding

This study was funded by the project of the Natural Science Foundation of Guangdong Province (2018A030313632), the Innovation and Entrepreneurship Training Program for College Students of GZUCM (S201910572049, 201910572168) and GZUCM Science Fund for Creative Research Groups (2016KYTD10) to Jin Zhang; the project of the Traditional Chinese Medicine Bureau of Guangdong Province (20211118) to Fuping Ding; and the Innovation and Entrepreneurship Training Program for College Students of GZUCM (201910572193) to Jin Huang. The funders had no role in study design, data collection and analysis, decision to publish, or preparation of the manuscript.

### Grant Disclosures

The following grant information was disclosed by the authors:
Natural Science Foundation of Guangdong Province: 2018A030313632.
Innovation and Entrepreneurship Training Program for College Students of GZUCM: S201910572049, 201910572168.
GZUCM Science Fund for Creative Research Groups: 2016KYTD10.
Traditional Chinese Medicine Bureau of Guangdong Province: 20211118.
Innovation and Entrepreneurship Training Program for College Students of GZUCM: 201910572193.

### Competing Interests

The authors declare there are no competing interests.

### Author Contributions

- Jiaying Liang, Yaofeng Zhi and Wenhui Deng conceived and designed the experiments, performed the experiments, analyzed the data, authored or reviewed drafts of the paper, and approved the final draft.
- Weige Zhou conceived and designed the experiments, performed the experiments, analyzed the data, prepared figures and/or tables, authored or reviewed drafts of the paper, and approved the final draft.
- Xuejun Li performed the experiments, analyzed the data, authored or reviewed drafts of the paper, and approved the final draft.
- Zheyou Cai performed the experiments, analyzed the data, prepared figures and/or tables, authored or reviewed drafts of the paper, and approved the final draft.
- Zhijian Zhu, Jinxiang Zeng, Wanlan Wu,Ying Dong analyzed the data, authored or reviewed drafts of the paper, and approved the final draft.
- Jin Huang and Yuzhuo Zhang performed the experiments, authored or reviewed drafts of the paper, and approved the final draft.
- Shichao Xu and Yixin Feng performed the experiments, authored or reviewed drafts of the paper, contributed reagents/materials/analysis tools, and approved the final draft.
- Fuping Ding and Jin Zhang conceived and designed the experiments, performed the experiments, authored or reviewed drafts of the paper, and approved the final draft.
## Data Availability

The data is available at NCBI GEO: GSE14520. All analyzed or generated data are included in the article and the Supplemental Files.

## Supplemental Information

Supplemental information for this article can be found online at http://dx.doi.org/10.7717/peerj.11627#supplemental-information.

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
