# Peer review of "Development and validation of ferroptosis-related lncRNAs signature for hepatocellular carcinoma"

_PeerJ, doi:10.7717/peerj.11627_

## Round 0.1 · original submission · Minor Revisions

Thank you for considering PeerJ for your manuscript submission titled “Development and validation of ferroptosis-related lncRNAs signature for hepatocellular carcinoma ”.

Take into account the comments of the three reviewers, I coincide with their opinions. While the referees do acknowledge that the manuscript has merit, the clear consensus is that modifications are needed. There are several topics remarked upon by the reviewers that should be addressed to improve the technical quality of your manuscript before submission of a revised version of your manuscript.

Therefore, I consider that the manuscript can be re-submitted as a revised version wherein provided that the recommendations are addressed to reinforce the biological phenomenon studied. Also, you should answer all questions and items requested point by point.
Thank you for your progress, and we look forward to receiving the revised manuscript.

Reviewer 1 ·

Basic reporting

Overall, the manuscript is well written. However, there are few minor issues that should be corrected

1. 1.Line 53: substitute “Immune checkpoint inhibitors and Kinase” by “Immune checkpoint inhibitors and kinase inhibitors” Do this if you refer also to inhibitors of kinases. Otherwise, the term “kinase” alone does not make sense.
2. Lines 65 - 66: substitute “defined as an autonomous” by “are autonomous”.
3. Lines 235 - 236: “Tumor immunotherapy utilizing ICB had gradually become a promising strategy for therapy of advanced HCC”. Please, add a proper reference supporting this statement
4. Lines 270 - 271: “ICB immunotherapy acted as emerging strategies has revealed treatment effects in HCC and other cancers”. Please, rephrase the sentence for a better understanding.
5. Line 293: Substitute “PART1 involved in cell migration and invasion” by “PART1 is involved in cell migration and invasion”
6. Line 299: substitute “possessing” “by possess”

Experimental design

No comment

Validity of the findings

Did the authors assess the probable influence of confounding factors like other comorbidities or alcohol consumption on both the robustness and accuracy of the FRlncRNAs signature? I consider that this should be assessed by virtue that other non - cancer liver diseases are also related with ferroptosis such as acute liver failure, alcoholic liver disease, NAFLD and immune-mediated hepatitis [Mao et al. Cell Death Dis. 2020;11(7):518]. Indeed, NAFLD has been related also with diabetes, insulin resistance, obesity, and overweight. Thus, the influence of these morbidities like confounding factors may be also tested.

Additional comments

It was found that the FRncRNAs signature may be a predictor of the overall survival of HCC patients; the higher the expression of FRncRNAs, the lower survival. Indeed, the score seems to be useful to measure the success of treatment with ICI. On the other hand, HCC can be resistant to conventional chemotherapeutic due to induction of ferroptosis resistance [Galmiche A. (2019) Ferroptosis in Liver Disease. In: Tang D. (eds) Ferroptosis in Health and Disease. Springer]. Does the FRncRNAs signature may be used to predict the resistance to classical chemotherapy via ferroptosis -induction? I wonder if ferroptosis may be easier to induce with conventional chemotherapy in patients with a higher FRncRNAs signature (i.e., more probability of liver ferroptosis induction) than in those with a lower one (i.e., lower probability of liver ferroptosis induction). I think that this may be an important question to address since ICI therapy is way expensive and has the risk of immune-related adverse events [Galle et al. J. Hepatol. 2018;69:182–236]. Please, discuss this possibility.

Reviewer 2 ·

Basic reporting

The paper has some grammatical errors

Experimental design

No comment

Validity of the findings

no comment

Additional comments

This study is aided to establish a ferroptosis-related lncRNAs (FRlncRNAs) signature in predicting the prognosis of patients with HCC. The work was well designed and written and the conclusions offered are consistent with the results. Although the work has several remaining issues as stated by the authors, it contains very valuable information related to FRncRNAs and its use as a signature of the patients with HCC, which could be used as a powerful tool in predicting the prognosis of patients with this type of cancer.


Minor points:

1.- Line 36, please add “overall survival” before “OS”.
2.- Line 139, change “were positive” with “had positive”.
3.- Line 222, please write “was implicated”.
4.- Line 247, delete “had”, in the same line, change “played” to “play”.
5.- Line 250, change “participated” to “participate”.
6.- Line 255, change “had” by “have”.
7.- Line 260, please change “this study existed several shortages” with “that study had several limitations”.
8.- Line 264, please remove “in”.
9.- Line 268 please change “might be served” with “might serve”.
10.- Line 270-271, please rewrite “acted as emerging strategies has revealed” to “emerged as a strategy to reveal”.
11.- Line 272, change “had” to “has”
12.- Line 273, change “response” to “respond”.
13.- Line 278, change “possessed” to “have”.
14.- Line 279, please rewrite “roles for measurement in response to ICB therapy” to “to be used to measure the response to ICB therapy”.
15.- Line 281, change “implied” to “implies”.
16.- Line 285, please change “has” to “have”.
17.- Line 292, change “had” to “has”.
18.- Line 299, change “possessing” to “possess”.
19.- Figure 8, panels A-F, a break in axis “X” at the Risk score value of 5 would help to amplify the plots to better appreciate the tendency of the data.
20.- Figure 6, please exchange the labels “D” and “C” in the panels.

Reviewer 3 ·

Basic reporting

no comment

Experimental design

1. the AUC of risk score in Figure 3B is different from that in figure 3D. Please improve it.
2. In multivariate Cox regression analysis, should be adjusted some risk factors, including family history, and Hepatitis C/B, which can be also obtained from the TCGA database.
3. Calibration curves should be consisted with the nomogram of 1-, 3-, and 5-year survival.
4. The cut-off of validation cohort should be provided reasonably .
5. The AUC of risk score for one, two, three in validation cohort. Please provide the AUC for one, three, five years.
6. Authors should investigate the effect of cross-talk between the risk scores and immune checkpoint genes on the prognosis.
7. The role and mechanism of ferroptosis and the ferroptosis-related genes in development and prognosis of HCC should be discussed.

Validity of the findings

please see 2.Experimental design.

Additional comments

MAJOR REVISIONS

---

## Round 0.2 · accepted · Accept

Thank you for considering PeerJ for your manuscript submission titled “Development and validation of ferroptosis-related lncRNAs signature for hepatocellular carcinoma”.

Taking into account the comments of the reviewers, I coincide with their opinions, the referees do acknowledge that the manuscript was substantially improved taken into account the reviewer's recommendations. Therefore, it could be accepted for publishing.

Sincerely